# Study on the Soil Microbial Diversity of *Cymbidium goeringii* and *Cymbidium faberi* in the Qinling Mountains after Introduction and Domestication

Ruixue Lv [1], Jing Zhang [2], Huimin Liao [3], Jean W. H. Yong [4] and Junyang Song [3,5,*]

1  College of Life Sciences, Nanjing University, Nanjing 210008, China; 502022300066@smail.nju.edu.cn
2  College of Horticulture, Northwest A&F University, Xianyang 712100, China; yyzhj@nwsuaf.edu.cn
3  College of Landscape Architecture and Art, Northwest A&F University, Xianyang 712100, China;
   liaohuimin19980213@163.com
4  Department of Biosystems and Technology, Swedish University of Agricultural Sciences,
   23422 Uppsala, Sweden; jean.yong@slu.se
5  Engineering and Technology Center for Breeding and Protection of Rare Plants in Qinba Mountains,
   Qinling Research Institute, Northwest A&F University, Xianyang 712100, China
*  Correspondence: songjunyang@nwafu.edu.cn; Tel.: +86-135-7240-9503

**Abstract:** Rhizosphere microbial communities have abundant species and a large number, and affect the physiology and growth of plants. When studying rhizosphere microbes, the rhizosphere ecosystem function and protection of wild orchids will be facilitated. By using high-throughput sequencing technology, the rhizosphere and non-rhizosphere bacteria and fungi of wild *Cymbidium goeringii* and *Cymbidium faberi* in the Qinling Mountains were analyzed at phylum, class, order, family, and genus levels to explore the rhizosphere bacterial and fungal community structure and diversity of orchid plants (*C. goeringii* and *C. faberi*) under natural conditions. The results showed that at the phylum level Proteobacteria was dominant in rhizosphere and non-rhizosphere soil of *C. goeringii* and *C. faberi*, but the proportion was different. The abundance of Proteobacteria in rhizosphere soil of *C. faberi* was the highest (35.5%), which was about 1.3 times of that in non-rhizosphere soil. Bacteroidetes accounted for 17.2% in rhizosphere soil of *C. goeringii*, much higher than that of non-rhizosphere soil (7.92%). The dominant groups of fungi in rhizosphere soil of *C. goeringii* and *C. faberi* were both Ascomycota. At the genus level, PCoA analysis showed that the community structure of bacteria and fungi in different samples was not only common but also specific, which was manifested in the similar dominant species but different subdominant species. This difference is reflected in the composition and relative abundance of microbial communities between different samples, and will gradually become obvious with the refinement of genera.

**Keywords:** *Cymbidium goeringii*; *Cymbidium faberi*; rhizosphere soil; microbial diversity; high-throughput sequencing

## 1. Introduction

The rhizosphere, which refers to the micro-region where soil–plant microorganisms act together to play an important role in the exchange of material between the plant root system and the soil and in the root system's own vital activities and metabolism, was introduced by German microbiologist Lorenz Hiltner in 1904 [1].

Due to the joint effort of plant metabolism and microbial activities, the characteristics and nutrients of rhizosphere soil are different from non-rhizosphere soil, and these changes in physical and chemical properties also alter the structure and composition of rhizosphere microbial communities [2–5]. Soil microbial diversity reflects the evolution of ecosystem functions and environmental stress, revealing differences in soil microbial species and in functions [6–8]. According to Xu et al. [9], Cavaglieri et al. [10], and Qin et al. [11], the structure and diversity of rhizosphere microbial communities are also affected by factors such as

plant variety, planting densities, and planting patterns. The structure and composition of rhizosphere microorganisms differ among various plant groups [12,13]. Some rhizosphere microorganisms are able to enhance plant diversity and adaptability through interactions with their hosts and are beneficial microorganisms that promote plant growth and development [14,15]; however, there are also harmful microorganisms that act as inhibitors, causing plant growth and development to be affected and disturbed [16]. Therefore, research on the diversity and interaction mechanisms of rhizosphere microorganisms is critical to the conservation and exploitation of plant resources [14].

Orchidaceae, as a comparatively evolved group of plants, is one of the most diverse families of angiosperms [17]. Traditional orchids generally refer to the native species produced in the south of the Qinling Mountains in China, of which *Cymbidium goeringii* and *Cymbidium faberi* are the most widely distributed [18].

*Cymbidium goeringii* is a common species of cymbidium, growing on rocky mountain slopes, forest margins, or in the light of forests at altitudes of 300 to 3000 m. It can be found in regions south of the Qinling Mountains in China, and also in India, Japan, and the southern part of the Korean Peninsula. Meanwhile, *Cymbidium faberi* grows in well-drained, moist, and transparent areas at altitudes of 400–3000 m; it can mainly be found in regions south of the Qinling Mountains in China, and also in Northeastern India and Nepal [19,20].

Based on the ecological functions, in this study the rhizosphere soil and non-rhizosphere soil of domesticated wild *Cymbidium goeringii* and *Cymbidium faberi* were collected from the Qinling Mountains in Shanxi province. The microbial community structure of rhizosphere soil of domesticated wild *Cymbidium goeringii* and *Cymbidium faberi* was analyzed by high-throughput sequencing technology, and the differences in microbial diversity and community structure in rhizosphere soil and non-rhizosphere soil were compared, mainly to understand the dominant fungi and bacteria in rhizosphere soil. It provides theoretical basis for the conservation, introduction, and domestication of *Cymbidium goeringii* and *Cymbidium faberi* in the wild.

## 2. Materials and Methods

### 2.1. Overview of the Experimental Site

The plant material for this experiment, *Cymbidium goeringii* and *Cymbidium faberi*, was collected in July 2008 from Zhen'an County, Shaanxi Province, China, and planted in the Orchid Garden of Northwest Agriculture and Forestry University, Yangling, China. Rhizosphere and non-rhizosphere soil samples were collected in July 2018 from the same Orchid Garden.

### 2.2. Methods

2.2.1. Soil Sampling Collection

According to the five-point sampling method, three *Cymbidium goeringii* and *Cymbidium faberi* plants were selected at each point, with a total of 15 plants of each orchid. The rhizosphere soil of every five plants was mixed into one sample, and three rhizosphere soil samples were obtained for *Cymbidium goeringii* and *Cymbidium faberi*. Rhizosphere soil samples were collected by first collecting soil about 1 cm above the surface as topsoil, digging up plants with intact soil between the roots, gently shaking off large pieces of soil that did not contain rhizosphere soil, and then brushing the soil attached around the roots into sterile bags with a sterile brush.

For non-rhizosphere soil, an area 5 m away from the rhizosphere soil collection site without any Orchidaceae was selected. The soil about 5 to 15 cm below the surface was taken, mixed, and placed into sterile bags for testing. The collected non-rhizosphere soils were thoroughly mixed and evenly divided into three parts.

2.2.2. Extraction of Soil Microbial DNA

A soil DNA extraction kit from MP Biomedicals was selected to test the total DNA extracted from the collected rhizosphere soil microorganisms of *Cymbidium goeringii* and

*Cymbidium faberi*. An ultramicro UV–Vis spectrophotometer was applied to measure the concentration and purity of the extracted DNA.

### 2.2.3. High-Throughput Sequencing

The DNA samples were stored at −80 °C and mailed to Shanghai Meiji Biomedical Technology Co for high-throughput sequencing.

The upstream primerITS1F(5′-CGTAGGTGAACCTGCGGAAGGATC-3′) and downstream primer ITS2R(5′-CTCGGAGGATCCTCGCC-3′) amplified fungi; the upstream primer515F(5′-GTGCCAGCMGCCGCGGTAA-3′) and downstream primer 907R(5′-CCGTC AATTCMTTTRAGTTT-3′) amplified bacteria. PCR amplification products were used for Illumina sequencing.

### 2.2.4. Data Processing and Analysis

The number of randomly selected sequences in the samples was used as the horizontal coordinate and the number of valid OTUs obtained by clustering based on this number of sequences was used as the vertical coordinate. Dilution curves were then plotted for the high-throughput sequencing results of rhizosphere soil samples and non-rhizosphere soil samples to show the validity of the number of samples sequenced and to compare the species' richness in samples with different amounts of sequencing.

The RDP database was used to construct sequencing libraries and process the data; the QIIME1.9.1 software was then applied to filter raw sequences and the UCHIME was used to check and remove chimera sequences [21]. All sequences were classified into OTUs based on their different similarity levels, and the RDP classifier Bayesian algorithm was applied to perform taxonomic analysis of the OTU representative sequences at the 97% similarity level; meanwhile, community composition was counted at each classification level, respectively: domain, kingdom, phylum, class, order, family, and genus. The silva138 database was used for bacteria 515f, and the UNITE8.0 database was used for fungi ITS.

## 3. Results

### *3.1. Sequencing Results and Quality Analysis*

#### 3.1.1. Bacteria

Through high-throughput sequencing, 22,430, 21,262, and 22,339 reads were obtained from the three *Cymbidium goeringii* rhizosphere soil samples CL1, CL2, and CL3, respectively; 22,430, 21,262, and 22,339 reads were obtained from the three *Cymbidium faberi* rhizosphere soil samples HL1, HL2, and HL3, respectively; and 18,689, 19,255, and 18,904 reads were obtained from the three non-rhizosphere soil samples KB1, KB2, and KB3, respectively. Based on the 97% similarity level, a total of 2455 reads were gathered as species analysis OTUs. A total of 2203, 2199, and 2188 OTUs were obtained from the *Cymbidium goeringii* and *Cymbidium faberi* rhizosphere soil bacteria and non-rhizosphere soil bacteria, respectively. The dilution curves were analyzed and the four parameters Ace, Chao, Simpson, and Shannon showed an increase as the number of reads grew, until reaching saturation (Figure 1), indicating that the depth of sequencing was sufficient for subsequent analysis of the rhizosphere bacterial community.

#### 3.1.2. Fungi

Through high-throughput sequencing, 68,095, 71,680, and 65,375 reads were obtained from the three *Cymbidium goeringii* rhizosphere soil samples CL1, CL2, and CL3, respectively; 70,993, 72,457, and 72,035 reads were obtained from the three *Cymbidium faberi* rhizosphere soil samples HL1, HL2, and HL3, respectively; and 67,336, 67,511, and 72,283 reads were obtained from the three non-rhizosphere soil samples KB1, KB2, and KB3, respectively. Based on the 97% similarity level, a total of 1168 reads were gathered as species analysis OTUs. A total of 912, 942, and 921 OTUs were obtained from the *Cymbidium goeringii* and *Cymbidium faberi* rhizosphere soil fungi and non-rhizosphere soil fungi, respectively. The dilution curves were analyzed and the four parameters Ace, Chao, Simpson, and Shannon

showed an increase as the number of reads grew, until reaching saturation (Figure 2), indicating that the depth of sequencing was sufficient for subsequent analysis of the rhizosphere fungal community.

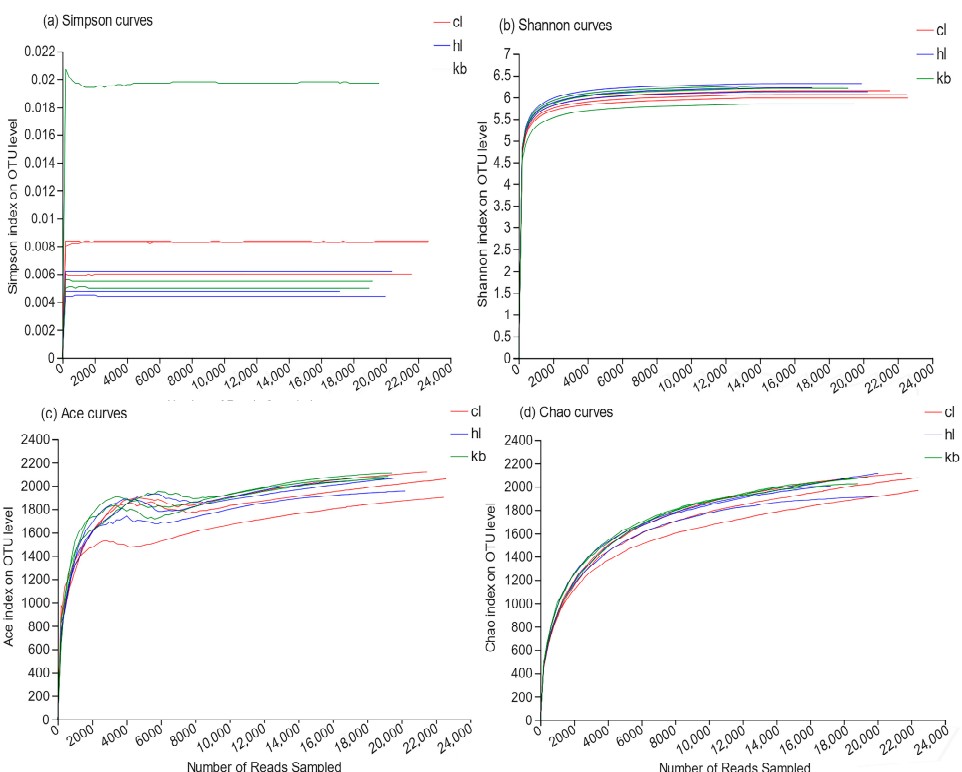

**Figure 1.** Bacterial dilution curves of samples (97%).

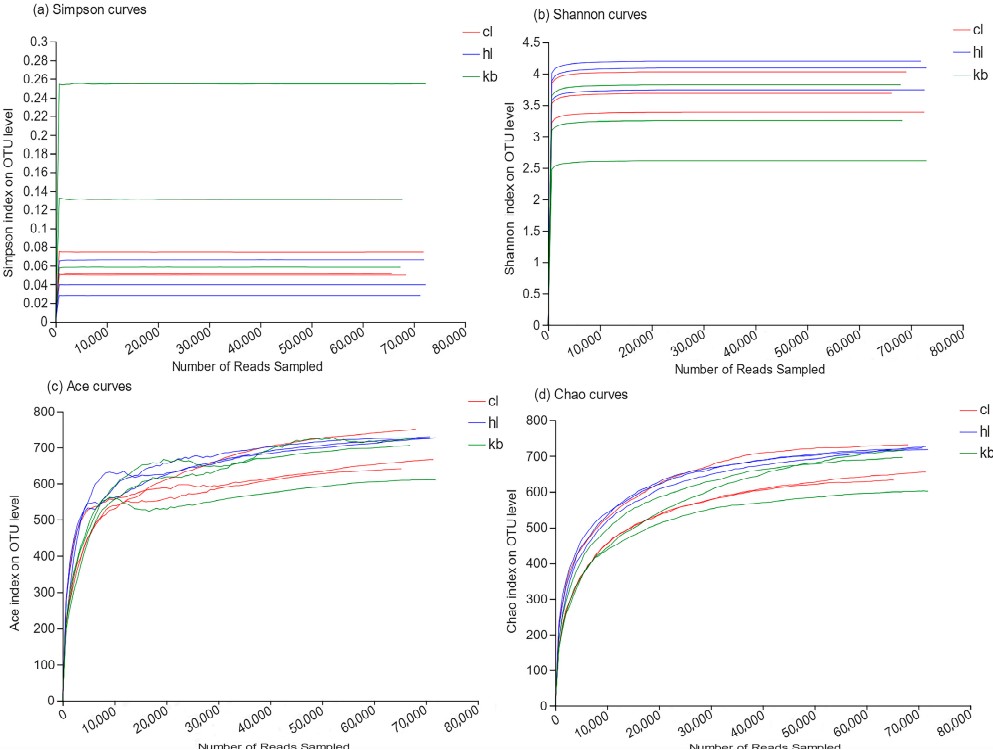

**Figure 2.** Fungal dilution curve of samples (97%).

### 3.2. Diversity Analysis of Bacteria in Rhizosphere

### 3.2.1. Analysis of Alpha Diversity in Bacteria

Bacterial Alpha diversity indexes were statistically analyzed for all soil samples (Table 1). The mean values of the Chao index/ACE index in *Cymbidium goeringii* and *Cymbidium faberi* rhizosphere soil samples and non-rhizosphere soil samples were 2052.12/2025.78, 2013.07/2021.16, and 2057.28/2081.12, respectively, indicating that the bacterial community richness of rhizosphere soil was higher. Meanwhile, in terms of bacterial community diversity, the Simpson index of rhizosphere soil was slightly lower than that of non-rhizosphere soil, indicating that the bacterial community diversity of *Cymbidium goeringii* and *Cymbidium faberi* rhizosphere soil was higher than that of non-rhizosphere soil. Combining the results of OTU counts and Alpha diversity analysis, it can be seen that the richness and diversity of bacterial communities in rhizosphere soil are higher than those in non-rhizosphere soil.

**Table 1.** Statistical results of bacterial diversity.

| Samples | Shannon | Simpson | Ace | Chao |
| --- | --- | --- | --- | --- |
| CL | 6.08 ± 0.06 a | 0.01 ± 0.00 a | 2025.78 ± 112.38 a | 2052.12 ± 74.68 a |
| HL | 6.22 ± 0.09 a | 0.01 ± 0.00 a | 2021.15 ± 55.70 a | 2013.07 ± 94.42 a |
| KB | 6.11 ± 0.22 a | 0.01 ± 0.01 a | 2081.12 ± 28.69 a | 2057.28 ± 29.12 a |

Different lowercase letters indicate the level of significant difference ($p < 0.01$).

The OTUs Venn diagram demonstrates the bacterial OTUs of *Cymbidium goeringii* (CL) and *Cymbidium faberi* (HL) rhizosphere soil samples and non-rhizosphere (KB) soil samples (Figure 3). A total of 2592 OTUs were obtained from the three sets of soil samples (CL, HL, and KB), with the number to CL being 2203, the number to HL being 2119, and the number to KB being 2188. In addition, the number specific to CL was 130, the number specific to HL was 137, and the number specific to KB was 66. A total of 277 OTUs were shared between CL and KB, 137 OTUs were shared between CL and HL, and 186 OTUs were shared between HL and KB, giving a total of 1659 OTUs shared for all three; the three soils had more of the same bacteria species. The OTU shared by the three is about 12.8 times that of CL, 12.1 times that of HL, and 25.1 times that of KB.

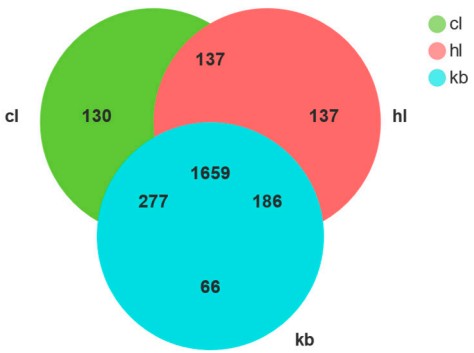

**Figure 3.** The OTU Venn graph of the sample bacterial.

The results suggest that a rich bacterial composition can be found in *Cymbidium goeringii* and *Cymbidium faberi* rhizosphere soil and non-rhizosphere soil. And, the three have many common microbiomes, indicating that the three microbiomes have little difference in the number and species of bacteria among the three varieties, among which the bacterial uniqueness was HL > CL > KB.

### 3.2.2. Analysis of Bacterial Taxonomic Composition

As can be seen in Figure 4a, a total of seven phyla are involved in the bacteria detected in the nine soil samples of *Cymbidium goeringii* and *Cymbidium faberi*, both rhizosphere and

non-rhizosphere. They are Proteobacteria, Acidobacteria, Bacteroidetes, Actinobacteria, Planctomycetes, Cyanobacteria, and Chloroflexi.

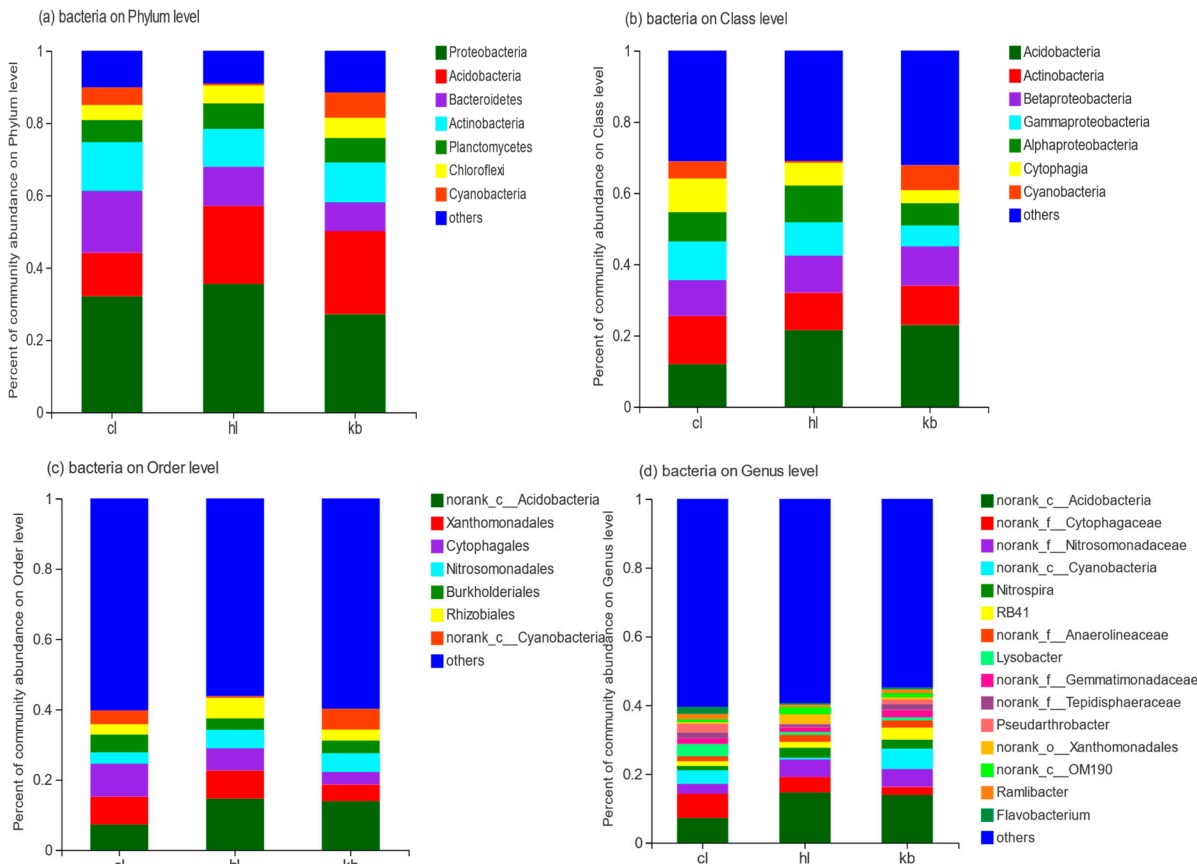

**Figure 4.** Horizontal distribution of bacteria. (**a**) analysis of the composition of bacteria at the phylum level; (**b**) analysis of the composition of bacteria at the class level; (**c**) analysis of the composition of bacteria at the order level; and (**d**) analysis of the composition analysis of bacteria at the genus level.

The dominant phylum of *Cymbidium goeringii* rhizosphere soil was Proteobacteria, accounting for 32.1%, about 1.2 times that of in non-rhizosphere soils. As for *Cymbidium faberi*, the dominant phyla were Proteobacteria and Acidobacteria (35.5% and 21.5%, respectively). Moreover, Bacteroidetes accounted for 17.2% of the bacteria in *Cymbidium goeringii* rhizosphere soil and 10.86% in *Cymbidium faberi* rhizosphere soil, much higher than that in non-rhizosphere soil. Actinobacteria accounted for 13.5% of the bacteria in the *Cymbidium goeringii* rhizosphere soil, which was greater in comparison to *Cymbidium faberi* rhizosphere soil and non-rhizosphere soil. On the other hand, the levels of Planctomycetes and Cyanobacteria in rhizosphere soil samples were not significantly different from those in non-rhizosphere soil samples. However, the relative richness of Cyanobacteria in *Cymbidium faberi* rhizosphere soil was quite low, being only at 0.51%. In addition, unknown bacteria also made up a certain proportion (about 10.2% in *Cymbidium goeringii* rhizosphere soil and 11.6% in *Cymbidium faberi* rhizosphere soil).

The dominant phylum of non-rhizosphere soil is Proteobacteria, which accounted for 27.1%. This was then followed by Acidobacteria (23.0%), Actinobacteria (11.0%), Bacteroidetes (8.0%), and Chloroflexi (5.0%). Meanwhile, 11.6% of bacteria in non-rhizosphere soil were unidentified.

As shown in Figure 4b, analysis of the composition of bacteria at the class level reveals that the seven major classes with high proportions in the three groups of soil samples are Acidobacteria, Acitinobacteria, Betaproteobacteria, Gammaproteobacteria, Alphaproteobacteria, Cytophagia, and Cyanobacteria. As for their richness, Acitinobacteria (12.0%) and

Gammaproteobacteria (10.8%) accounted for a higher proportion in Cymbidium goeringii rhizosphere soil samples, while in *Cymbidium faberi* rhizosphere soil samples and non-rhizosphere soil samples, the dominant class was Acidobacteria (21.5% for *Cymbidium faberi* rhizosphere soil samples, and 23.0% for non-rhizosphere soil samples).

As shown in Figure 4c, analysis of the composition of bacteria at the order level shows that the seven major orders with high proportions in the three groups of soil samples are norank_c_Acidobacteria, Cytophagales, Xanthomonadales, norank_c_Cyanobacteria, Burkholderiales, and Nitrosomonadales. The dominant order of *Cymbidium goeringii* rhizosphere soil samples was Cytophagates (9.4%), which accounted for 3.6% in the non-rhizosphere soil samples, while the dominant order of *Cymbidium faberi* rhizosphere soil samples was norank_c_Acidobacteria (14.6%). Moreover, Xanthomonadales accounted for 7.9% of both *Cymbidium goeringii* and *Cymbidium faberi* rhizosphere soil, which was much higher than that of non-rhizosphere soil (4.7%). However, Nitrosomonadales and Burkholderiales were less represented, while norank_c_Cyanobacteria and Burkholderiales were more evenly represented in the samples, at around 4% to 6%.

As shown in Figure 4d, analysis of the composition of bacteria at the genus level illustrates that there are significant differences in the composition of *Cymbidium goeringii* rhizosphere soil bacteria and non-rhizosphere soil bacteria. The more abundant of both were genera that could not be identified. Of the genera that could be identified, *norank_c_Acidobacteria* was the dominant genus in both *Cymbidium goeringii* and *Cymbidium faberi* rhizosphere soil samples (7.22% for *Cymbidium goeringii* and 14.6% for *Cymbidium fiaberi*). The proportion of *norank_c_Cyanobacteria* in *Cymbidium goeringii* rhizosphere soil was less than that in non-rhizosphere soil: 3.93% and 5.92%, respectively. However, the percentage of *norank_f_Nitrosomonadaceaed* in non-rhizosphere soil was 5.23%, which was greater than that in *Cymbidium goeringii* rhizosphere soil (2.90%). In contrast, *norank_f_Cytophagaceae* accounted for 7.05% of *Cymbidium goeringii* rhizosphere soil, a much higher percentage than the 2.29% of non-rhizosphere soil. Meanwhile, the proportion of *Lysobacter* was 1.39% in *Cymbidium goeringii rhizosphere* soil compared with only 0.83% in non-rhizosphere soil. There were more *norank_c_Cyanobacteria* in both *Cymbidium goeringii* rhizosphere soil and non-rhizosphere soils in comparison with *Cymbidium fiaberi* rhizosphere soil (3.0%).

### 3.3. Diversity Analysis of Fungi in Rhizosphere Soil

3.3.1. Analysis of Alpha Diversity in Fungi

Fungi Alpha diversity indexes were statistically analyzed for all soil samples (Table 2). The mean values of the Chao index/ACE index in *Cymbidium goeringii* and *Cymbidium faberi* rhizosphere soil samples and non-rhizosphere soil samples were 672.74/686.73, 721. 81/727.68, and 671.82/681.04, respectively, indicating that the fungi community richness of rhizosphere soil was higher. The Simpson index of the rhizosphere soil was much lower than that of non-rhizosphere soil, suggesting that more diverse fungal communities in rhizosphere soil can be detected. Thus, taking both the OTU counts and Alpha diversity results into consideration, it can be concluded that the fungal community richness and diversity of the rhizosphere soil is higher than that of non-rhizosphere soil.

**Table 2.** Statistical results of fungi diversity.

| Samples | Shannon | Simpson | Ace | Chao |
|---|---|---|---|---|
| CL | 3.71 ± 0.32 a | 0.06 ± 0.01 a | 686.73 ± 57.62 a | 672.74 ± 51.85 a |
| HL | 4.02 ± 0.24 a | 0.04 ± 0.02 a | 727.68 ± 2.47 a | 721.81 ± 3.36 a |
| KB | 3.24 ± 0.60 a | 0.15 ± 0.10 a | 681.04 ± 61.81 a | 671.82 ± 60.32 a |

Different lowercase letters indicate the level of significant difference ($p < 0.01$).

The OTUs Venn diagram displays the fungal OTUs of *Cymbidium goeringii* (CL) and *Cymbidium faberi* (HL) rhizosphere soil samples and non-rhizosphere (KB) soil samples (Figure 5). A total of 1423 OTUs were obtained from the three sets of soil samples (CL, HL, and KB), with the number to CL being 912, the number to HL being 942, and the number

to KB being 921. In addition, the number specific to CL was 119, the number specific to HL was 255, and the number specific to KB was 158. Meanwhile, 204 OTUs were shared between CL and KB, 128 OTUs were shared between CL and HL, and 98 OTUs were shared between HL and KB, giving a total of 461 OTUs shared for all three. The OTU shared by the three is about 3.9 times that of CL, 1.8 times that of HL, and 2.9 times that of KB.

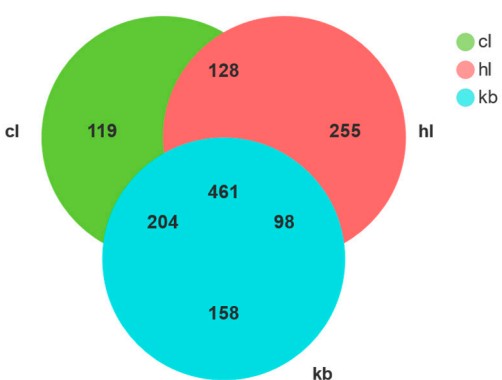

**Figure 5.** The OTU Venn graph of the sample fungi.

The results suggest that a rich fungal composition can be found in *Cymbidium goeringii* and *Cymbidium faberi* rhizosphere soil and non-rhizosphere soil. And, the three have many common microbiomes, indicating that the three microbiomes have little difference in the number and species of fungal among the three varieties, among which the fungal uniqueness was HL > KB > CL.

### 3.3.2. Analysis of Fungi Taxonomic Composition

As shown in Figure 6a, a total of five phyla are involved in the fungi detected in the nine soil samples of *Cymbidium goeringii* and *Cymbidium faberi*, both rhizosphere and non-rhizosphere. They are Ascomycota, Basidiomycota, unclassified_k_Fungi, Zygomycota, and Glomeromycota.

The dominant phylum of *Cymbidium goeringii* and *Cymbidium faberi* rhizosphere soil and non-rhizosphere soil was the same: Ascomycota (77.84%, 80.91%, and 79.80%, respectively). Furthermore, Zygomycota accounted for 6.56% of the fungi in *Cymbidium goeringii* rhizosphere soil, which was almost 2.4 times that of non-rhizosphere soil. Meanwhile, the amount of Basidiomycota in *Cymbidium faberi* rhizosphere soil (12.73%) was greater than that in both *Cymbidium goeringii* rhizosphere soil and non-rhizosphere soil.

As shown in Figure 6b, analysis of the composition of fungi at the class level reveals that the six major classes with high proportions in the three groups of soil samples are Sordariomycetes, Dothideomycetes, unclassified_k_Fungi, norank_p_Zygomycota, Agaricomycetes, and Tremellomycetes. The content of Sordariomycetes in all three sets of soil samples was high and basically the same, at 58.59%, 61.12%, and 59.87%, respectively. Furthermore, less Dothideomycetes and much less Agaricomycetes were found in *Cymbidium goeringii* rhizosphere soil compared with in *Cymbidium faberi* rhizosphere soil and non-rhizosphere soil.

According to Figure 6c, analysis of the composition of fungi at the order level indicates that the nine major orders with high proportions in the three groups of soil samples are Hypocreatess, norank_c_Sordariomycetes, Xylariale, Sordariates, unclassified_k_Fungi, Capnodiates, PleosPorates, unclassified_c_Sordariomycetes, and Sebacinales. Hypocreates had the highest proportion in all samples (24.0% for *Cymbidium goeringii* rhizosphere soil, 37.81% for *Cymbidium faberi* rhizosphere soil, and 32.29% for non-rhizosphere soil) among these. In addition, the dominant orders of *Cymbidium goeringii* rhizosphere soil samples were Xylariales (13.6%) and Sordariates (11.4%), and for *Cymbidium faberi* rhizosphere soil, Sordariates (10.10%) and PleosPorates (9.85%) were the relatively abundant orders. On the other side,

6.25% Sebacinales was detected in non-rhizosphere soil, while it was barely found in both *Cymbidium goeringii* and *Cymbidium faberi* rhizosphere soil.

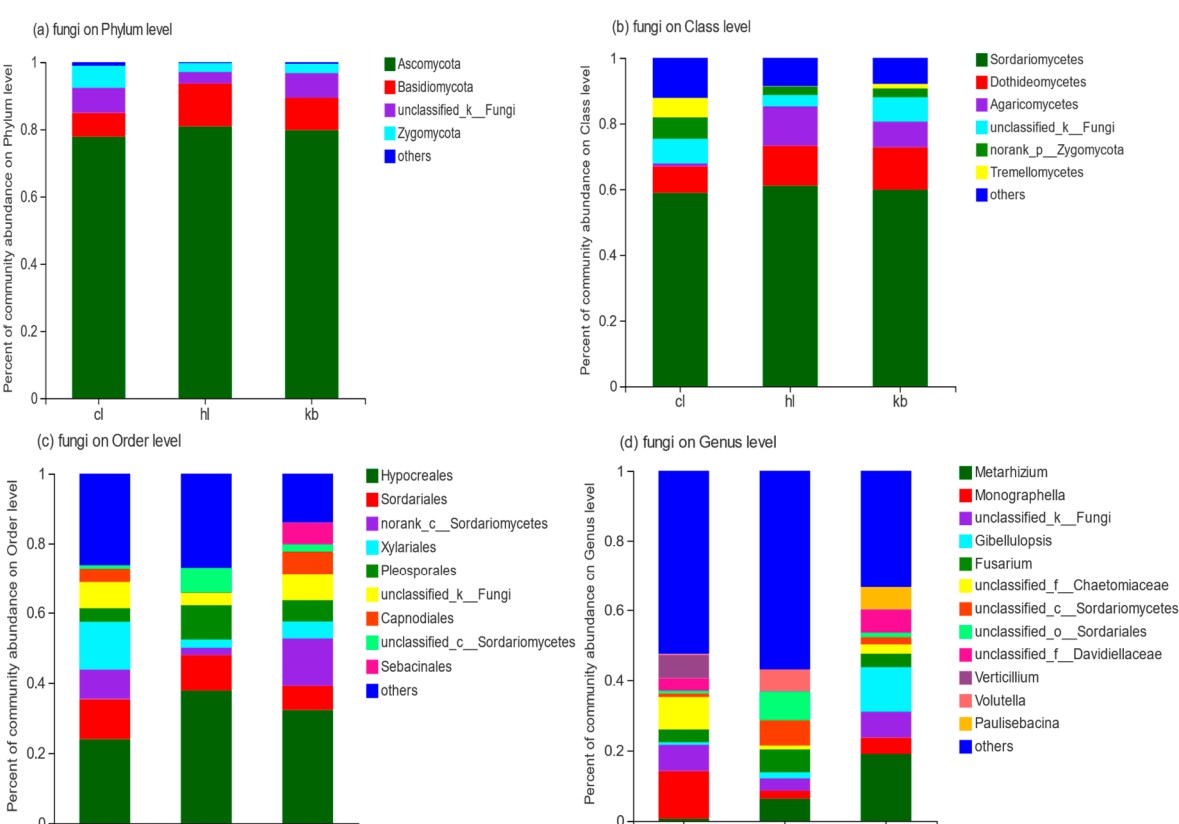

**Figure 6.** Horizontal distribution of fungi. (**a**) analysis of the composition of fungi at the phylum level; (**b**) analysis of the composition of fungi at the class level; (**c**) analysis of the composition of fungi at the order level; and (**d**) analysis of the composition analysis of fungi at the genus level.

Analysis of the composition of fungi at the genus level is shown in Figure 6d. As can be seen, the fungi detected in all three sets of soil samples contained a large number of genera, all showing great variability and diversity. *Monographella* (13.5%) was the dominant genus in *Cymbidium goeringii* rhizosphere soil, while it presented only 4.6% in non-rhizosphere soil. As for *Cymbidium faberi* rhizosphere soil, the content of *unclassified_o_Sordariales* and *unclassified_c_Sordariomycetes* had a greater proportion. Additionally, the content of *Verticillium* (6.66% in *Cymbidium goeringii* rhizosphere soil) and *Volutella* (6.11% in *Cymbidium faberi* rhizosphere soil) were both found to account less than 0.2% in non-rhizosphere soil. In contrast, *Gibellulopsis* was present in significant amounts (12.64%) in non-rhizosphere soil, but barely found in *Cymbidium goeringii* and *Cymbidium faberi* rhizosphere soil.

### 3.4. Analysis of Differences in Bacterial and Fungal Communities

To observe the variability of bacterial and fungal communities between samples, nine soil samples were analyzed at the genus level by principal coordinate analysis (PCoA). As shown in the PCoA analysis of the bacterial community (Figure 7a), PC1 and PC2 could explain 49.45% and 22.27% of all variables, respectively, with a cumulative contribution rate of 71.72%. In addition, the PCoA analysis of the fungal community (Figure 7b) indicates that PC1 and PC2 could explain 34.49% and 16.57% of all variables, respectively, with a cumulative contribution rate of 51.06%. It is apparent that these outcomes of PCoA analysis could characterize the composition of the microbial communities. Further analysis suggested that the bacterial communities in rhizosphere soil samples differed significantly

at the genus level from those in non-rhizosphere soil samples. In addition, compared with the fungal communities in *Cymbidium faberi* rhizosphere soil samples, there was some similarity between the fungal communities in *Cymbidium goeringii* rhizosphere soil and non-rhizosphere soil, while tremendous differences can be found between the fungal communities in *Cymbidium faberi* rhizosphere soil and non-rhizosphere soil. To be more specific, there were also distinctions between the composition of fungal communities in *Cymbidium goeringii* rhizosphere soil samples and *Cymbidium faberi* rhizosphere soil samples.

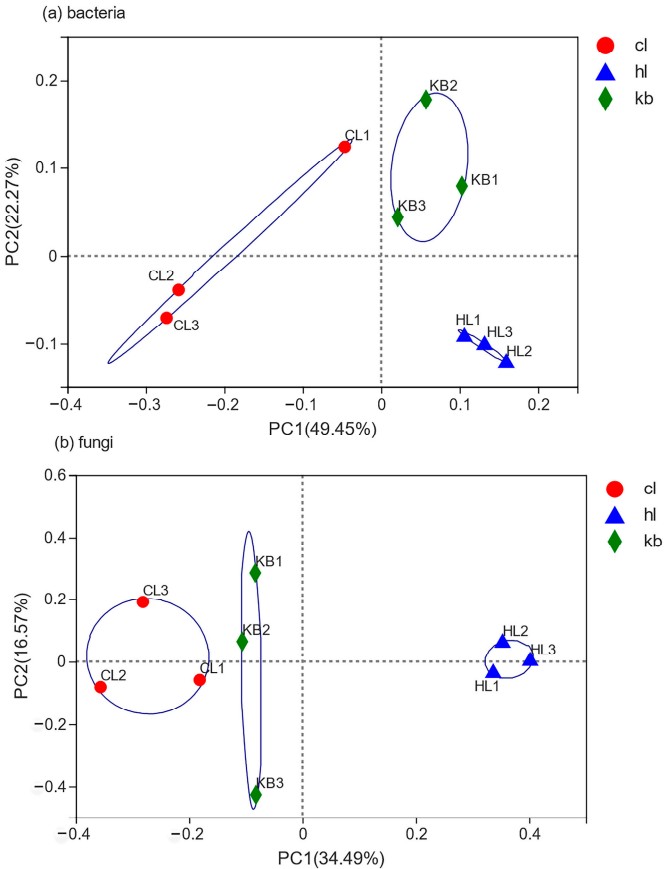

**Figure 7.** Principal component analysis of microorganism communities between different samples.

## 4. Discussion

In this paper, high-throughput sequencing technology was used to identify bacterial and fungal species in rhizosphere soil and non-rhizosphere soil of wild *Cymbidium goeringii* and *Cymbidium faberi* and to analyze microbial community structure and species diversity. It was also used to provide theoretical data support for the future introduction and domestication of wild orchids, and for substrate research and microbial fertilizer research, which are of great significance for the microhabitat research and resource protection of wild orchids. There is a complex interdependent and interactive relationship between plant and soil microorganisms. The abundant microorganisms in the microenvironment of rhizosphere soil, which include bacteria, fungi, and actinomycetes, are not only numerous but also diverse [22]. The biomass of rhizosphere microorganisms is roughly ranked as bacteria > actinomycetes > fungi, where there are both beneficial and harmful microorganisms that promote or inhibit plant growth and development. Present research revealed both similarities and differences in the rhizosphere bacteria and fungi of *Cymbidium goeringii* and *Cymbidium faberi*. This variation is reflected in the types, contents, and dominant species of bacteria and fungi between the two samples. Furthermore, it varies according to classification levels, as the variation tends to become more pronounced as the classification level becomes more refined, with the phylum level < the class level < the order level < the genus

level. Consequently, research on the diversity and interaction mechanisms of rhizosphere microorganisms plays a critical role in the conservation and exploitation of high-quality plant resources.

The rate of metabolism and reproduction of bacteria results in them being the most abundant group of plant rhizosphere microorganisms. Proteobacteria are the dominant rhizosphere bacteria in both *Cymbidium goeringii* and *Cymbidium faberi*, accounting for more than 30%. The relative richness of Cyanobacteria in the *Cymbidium faberi* rhizosphere soil was quite low (at 0.51%), much lower than in non-rhizosphere soil. It has also been shown that Alphaproteobacteria and Gammaproteobacteria, which are usually enriched in plants' environment, have essential interactions with plant growth and development. They contain a large number of total transfer proteins (including ATP-binding cassettes, phosphotransferase systems, and drug/metabolite transfer proteins) and can import or export a variety of compounds [22,23]. The common bacterial taxa found in plant rhizosphere are *Pseudomonas* spp., *Flavobacterium* spp., Alkali-producing *Bacillus* spp., *Chromobacterium* spp., and *Colorless Bacillus* spp. [24]. In fact, according to Prashar et al. (2013) [25], rhizosphere bacteria can be classified into three main categories: beneficial, neutral, and harmful. Beneficial rhizosphere bacteria, also known as plant growth-promoting bacteria (PGPR), is a general term for a group of bacteria that are colonized in and closely related to the root system, and they may promote plant growth directly or indirectly [26]. Several studies reveal that there are numerous bacterial communities in the rhizosphere that are beneficial to plants. Silva et al. (2004) [27] found that a variety of rhizosphere bacteria could promote plant growth directly or indirectly through the production of plant growth substances, antibiotics, and siderophore. Furthermore, there are complex interactions between rhizosphere bacteria and plants, and plenty of previous studies have identified beneficial rhizosphere bacteria that have antagonistic or pro-growth effects on plants. In other words, these bacteria can lower the use of pesticides and fertilizers and thus play a helpful part in promoting plant growth and reducing environmental pollution. Present research suggested that the dominant group of soil bacteria in both *Cymbidium goeringii* rhizosphere soil and *Cymbidium faberi* rhizosphere soil was Proteobacteria, followed by Bacteroidetes, and the content of them in rhizosphere soil was much higher than in non-rhizosphere soil.

Fungi are widely distributed in rhizosphere soils, especially in relatively impoverished soils, with seasonal variations in community structure. Research shows that higher fungal biodiversity in rhizosphere soil can be found than in non-rhizosphere soil. They are a vital part of the soil microbiota and play an indispensable role in the balance of soil microbiota. Oros-Sichler et al. studied the biodiversity of rhizosphere fungal communities in soil using DGGE fingerprinting, and suggested that soil had a greater effect on rhizosphere fungi biodiversity than soil type did [28]. The dominant groups of fungi in *Cymbidium goeringii* and *Cymbidium faberi* rhizosphere soil in this study were both Ascomycota. On the other hand, Verticillium and Volutella were detected in the rhizosphere fungi of both *Cymbidium goeringii* and *Cymbidium faberi*, while almost none were found in the non-rhizosphere soil. However, Gibellulopsis was present in large amounts (12.64%) in the non-rhizosphere soil, while very little was detected in the two rhizosphere soil samples. It has been indicated that there were 35 genera of symbiotic fungi for orchidaceae, belonging to 12 orders and 22 families of Basidiomycota and Ascomycota, respectively, with rhizoctonia solani predominant [29–31]. During the growth and development of orchidaceae, rhizosphere fungi play the part of decomposing organic matter, providing nutrients, promoting the absorption of trace element, and promoting seed germination as well as seedling establishment [32].

In addition, plants affect rhizosphere microorganisms significantly in two ways. To begin with, for different plant species the specific elements of rhizosphere microflora may vary, some of which are more beneficial to the establishment of rhizosphere microbial communities for plant growth [33,34]. On top of that, root exudates also impact the community and abundance of rhizosphere microorganisms [34,35]. Ladygina and Hedlund (2010) [36] applied a combination of $^{13}C$ isotope labeling and a phospholipid fatty acid to investigate

the rhizosphere microorganism communities in Lotus corniculatus, Plantago lanceolata, Holcus lanatus, and their mixed planting area. Their results revealed that more $^{13}$C markers could be found to enter the rhizosphere bacteria and actinomycetes in Holcus lanatus and the mixed planting area. The effect of root exudates on rhizosphere microorganisms also varies from plant to plant. They are intermediates in the interaction between plants and their rhizosphere microorganisms, with some attracting specific microbial groups and some inhibiting certain microbial groups [37]. The root exudates of various plants or the same plant in different development stages may be diverse, which leads to distinct rhizosphere microorganisms. Take Chiellini et al. (2014) [38] as an example; the microbial communities they found in Echinacea purpurea and Echinacea angustifolia planted in the same soil were quite different.

Orchidaceae is one of the most highly evolved groups of angiosperm plants, and orchids are not only of great scientific value, but are also extremely cherished for ornamental and medicinal purposes [39]. Due to the difficulty of reproduction under natural conditions, the seed germination rate and reproduction rate of Orchidaceae is extremely low. Moreover, long-term asexual reproduction has resulted in an increase in the number of virus-carrying plants. These natural limitations have led to slow progress in scientific research on Orchidaceae and the development of the flower industry. Some studies found that Orchidaceae can form mycorrhizal association with fungi in their habitat, and that those orchid mycorrhizal fungi (OMF) could promote seed germination and formation of protocorm. Some RAB (Root Associated Bacteria) can also promote growth and development of Orchidaceae [40–42].

In addition, rhizosphere microorganisms are highly diverse and specific, and may change during different life cycles of Orchidaceae [42,43], e.g., in most cases, OMF diversity tends to decline from seed to seedling period and often increases again in an hourglass-like pattern during the mature period. The composition and richness of Orchidaceae rhizosphere microorganisms can influence the adaptation and distribution of these plants [44]. And, it has been shown that OMF is unevenly distributed in small patches in the soil, which may be related to the clustered patchy distribution of Orchidaceae. Yet, its distribution pattern in the soil remains unclear [21,45].

The generality of rhizosphere bacteria over common soil bacteria is greater than the difference, probably due to the nature and role of the bacteria, whereas the difference between rhizosphere fungi and common soil fungi is much greater than the generality. This may be related to the functions of fungi and root exudates, and the material exchange among fungi, plants, and soil. Both Cymbidium goeringii and Cymbidium faberi are fleshy roots, and they must rely on microorganisms for root absorption. The interactions between Orchidaceae and their rhizosphere microorganisms are complex, and it is advisable to screen beneficial microbial resources in the future study so as to provide references for the cultivation and exploitation of Orchidaceae.

**Author Contributions:** J.S.: conceptualization, methodology, investigation, and resources; R.L.: data curation, formal analysis, and writing—original draft preparation; J.Z. and H.L.: writing—review and editing; J.W.H.Y.: investigation and writing—review and editing. All authors have read and agreed to the published version of the manuscript.

**Funding:** This research was funded by the National Forestry and Grassland Administration of China, grant number KJZXSA202114; the National Forestry and Grassland Administration of China, grant number 2020070713; the Department of Science and Technology of Shaanxi Province, grant number 2021NY-055; and the Forestry Bureau of Shaanxi Province, grant number SXLK2021-0211.

**Institutional Review Board Statement:** Not applicable.

**Data Availability Statement:** The data presented in this study are available under permission from the corresponding author upon reasonable request.

**Acknowledgments:** All the experiments in this paper were completed in Northwest A&F University. We would like to express our gratitude to Northwest A&F University.

**Conflicts of Interest:** There are no conflict of interest. The funders had no role in the design of this study; in the collection, analyses, or interpretation of data; in the writing of the manuscript; or in the decision to publish the results.

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
