# Peer review of "Study on the Soil Microbial Diversity of Cymbidium goeringii and Cymbidium faberi in the Qinling Mountains after Introduction and Domestication"

_diversity, doi:10.3390/d15090951_

Round 1

Author Response

Point 1: Line 26: i consider irrelevant that the second class of fungi was “unclasssified”. Either

you mention the third class or you just leave the Ascomycota with the percentage.

Line 27: unclassiied _k_Fungi. -> unclassified

Line 28: please rewrite the phrase as you are confusing the reader…with mentions on

the which variable explained higher variance, for instance..

Response 1: The reference to "unclassified fungi" has been removed from the abstract, and references to "unclassiied _k_Fungi" have been revised to "unclassified fungi" in the body.

Point 2: what are the objectives of this study?

Response 2: In this study, the rhizosphere soil and non-rhizosphere soil of domesticated wild Cymbidium goeringii and Cymbidium faberi were collected from Qinling Mountains in Shaanxi province. The microbial community structure of rhizosphere soil of domesticated wild Cymbidium goeringii and Cymbidium faberi was analyzed by high-throughput sequencing technology, and the differences of microbial diversity and community structure in rhizosphere soil and non-rhizosphere soil were compared, mainly to understand the dominant fungi and bacteria in rhizosphere soil. It provides theoretical basis for the conservation, habitat, substrate, microbial fertilizer and introduction and domestication of Cymbidium goeringii and Cymbidium faberi.

Point 3: Line 79: According to the five-point sampling method, three Cymbidium goeringii and

Cymbidium faberi, plants were selected at each point, with a total of 15 plants of each orchid. -> you mean by that 5 replicates with 3 plant individuals of each orchid species?

Line 89: The collected rhizosphere soil and non-rhizosphere soi were thoroughly mixed and 89 evenly divided into three parts. Each set of samples consisted of three biological replicates. -> please explain clearly your field assay from the plant individual grouped into..up to the final samples from which you have extracted the total DNA. By far I understand that you have 9 samples (i.e. dna, pcr and seq reactions): 3 for orchid species (rhizo soil) 1, 3 for orchid species 2 (rhizo soil) and 3 for non-rhizopshere soil. But the paragraph on field design for soil sampling is confusing…in my opinión you should explain it clearly.

Response 3: At present, there are 9 samples, which are respectively Cymbidium goeringii rhizosphere soil, Cymbidium faberi rhizosphere soil and non-rhizosphere soil. According to the five-point sampling method, three Cymbidium goeringii and Cymbidium faberi, plants were selected at each point, with a total of 15 plants of each orchid. The rhizosphere soil of every five plants was mixed into one sample, and three rhizosphere soil samples were obtained for Cymbidium goeringii and Cymbidium faberi. For non-rhizosphere soil, an area 5 m away from the rhizosphere soil collection site without any Orchidaceae was selected. The soil about 5 to 15cm below the surface was taken, mixed, and placed into sterile bags for testing. The collected non-rhizosphere soi were thoroughly mixed and evenly divided into three parts.

Point 4: Lin e96: what primers did you use for bacterial and fungal communities?Line 97: please specifiy the type of sequencing machine (i.e. Illumina technique, nanopore seq etc.)Line 107: QIIME software -> which versión?Line 110: 97% similarity level -> for which level of taxonomy? You should have used different thresholds for different levels…But please specify that this value was used for all taxonomical ranks, if so..Line 113: UNITE database was used for fungi ITS. -> which version?

Response 4: Use upstream primerITS1F(5’-CGTAGGTGAACCTGCGGAAGGATC-3’), down-stream primer ITS2R(5’-CTCGGAGGATCCTCGCC-3’) amplified fungi, upstream pri-mer515F(5’-GTGCCAGCMGCCGCGGTAA-3’), downstream primer 907R(5’-CCGTCAATTCMTTTRAGTTT-3’) amplified bacteria. PCR amplification prod-ucts were used for Illumina sequencing.

Version information:QIIME1.9.1 、silva128、UNITE8.0

Analysis parameter:Cluster mode :USEARCH7-uparse algorithm,OTU sequence similarity :0.97,Classification confidence :0.7,Species classification database :silva128/16s bacteria

Point 5: Line 158: OUTs-> please check trhought the document as it is mispelled multiple

times.

Response 5: The spelling of "OTU" has been rechecked and changed.

Point 6: The OTUs-Venn diagram demonstrated the bacterial OUTs of Cymbidium goeringii 158 (CL) and Cymbidium faberi (HL) rhizosphere soil samples and nonrhizosphere (KB) soil 159 samples (Figure 3). -> what did the venn diagram demonstrated? Also, please mention that there were no statistical differences between Cymbidium goeringii (CL), Cymbidium faberi rhizosphere soil samples and non-rhizosphere (KB) in terms of bacterial diversity índices. Please indicate the test for p value (tukey?).

Response 6: The bacterial OTU numbers of Cymbidium goeringii rhizosphere soil (CL), Cymbidium faberi rhizosphere soil (HL) and non-rhizosphere soil (KB) shown in the venn diagram reflect the number of unique and common bacterial species of the three, and a certain analysis has been made on this. The section on statistical differences between rhizosphere soil samples and non-rhizosphere soil samples has been added, and the section on fungi below has also been added in response.

Point 7: Figure 7. The samples based on horizontal principal coordinate analysis. -> the legends should be self explanatory. Please carefully rewrite the legends.

Response 7: The legend has been modified.“Principal component analysis of microorganism communities between different samples

Point 8: Line 337: how about studies on orchid bacterial communities? Please mention some relevant results to your study… please develo pon your findings…it is normal to observe these taxons around orchids? First question should be..and so on…Line 366: you talk about other studies with mention so families and orders but then you get straight to a species? In your study you have identified at genus level..Did you find this genus (i.e. Rhizoctonia) in our samples? Mention anything about the the things in common and the differences between your results and others…

Line 373-388: please resume in one frase and don´t forget your red thread of the reasoning…! This is totally irrelevant.. More, you don´t even compare or discuss your data, you just lay on some information without connecting it to your study…Line 390-399: move it to the intriduction chapter unless you do something with this information, regarding your data, of course…Line 401-407: similar to the above considerationLine 409: you are not saying anything by stating “Present research revealed both similarities and differences in the rhizosphere bacteria 409 and fungi of Cymbidium goeringii and Cymbidium faberi.”

Response8: In the discussion part, I added some content, combining my research purpose and results with others' research to compare and discuss.

Reviewer 2 Report

See Remarks

Author Response

Point 1: The objectives of the study carried out must be specified;

Response 1: The objective of this study was to understand the microbial community structure of rhizosphere soil and non-rhizosphere soil of Cymbidium goeringii and Cymbidium faberi in Qinling Mountains after domestication, and to compare the diversity and species of soil microbial communities in different ecological niches, so as to provide theoretical data support for the future introduction and domestication of wild orchids, substrate research and microbial fertilizer research, which is of great significance for the microhabitat research and resource protection of wild orchids.

Point 2: Discuss the presence of Volutela and Verticillium in the rhizosphere of the plant species

studied. Some representatives of these two species are true plant pathogens. One can also ask

the question about the origin of Verticillium in these areas which are not cultivable;

Response 2: The comparison of species and diversity of fungi and bacteria in the rhizosphere compared with non-rhizosphere soil was mainly discussed. The growth of wild Cymbidium goeringii and Cymbidium faberi after transplanting and domestication was slower than that in the wild habitat, but the plants were healthier, so the pathogens in the rhizosphere were not specifically discussed.

Point 3: Discuss how the microflora detected in the rhizosphere can be valorized in the form of

inocula capable of stimulating plant growth or protecting their roots against pathogens:

isolation, multiplication and formulation;

Response 3: Orchids cannot grow without microorganisms. According to the survey on the growth of orchids in the wild and domesticated areas, orchids grow better in the wild, so it is very important to detect microorganisms that can promote the growth of orchids. This paper also discusses some plant growth promoting bacteria.

Point 4: Also discuss how the microflora encountered (bacteria and fungi) can interact with the endophyte microflora of the plant species studied.

Response 4: Both plant endophytic microbiota and soil microbiota are very important to plants. This paper mainly studies soil microbiota, and briefly discusses the symbiotic microbiota of orchidaceae and their interaction.

Round 2

Reviewer 1 Report

Dear Authors, 

Pleake make the following small modifications: 

Line 27: PCoA analysis showed that fungal and bacterial species were specific to soil samples in both Cymbidium plant species.

Line 274: singular fungal species were found with higher values in HL than KB and CL. 

Author Response

Point 1: Line 27: PCoA analysis showed that fungal and bacterial species were specific to soil samples in both Cymbidium plant species.

Response 1: The original text is“At genus level, PCoA analysis showed that the community structure of bacteria and fungi in different samples was not only common but also different, which was mani-fested in the similar dominant species but different subdominant species.Differentones have been modified for “specific.

Point 2: Line 274: singular fungal species were found with higher values in HL than KB and CL.

Response 2: “among which the bacteria uniqueness was HL﹥KB﹥CL” be changed to“among which the fungal uniqueness was HL﹥KB﹥CL”. Represents the number of fungal OTUs specific to the three samples compared.
